# Comparative Proteomic Analysis Identifies EphA2 as a Specific Cell Surface Marker for Wharton’s Jelly-Derived Mesenchymal Stem Cells

**DOI:** 10.3390/ijms21176437

**Published:** 2020-09-03

**Authors:** Ashraf Al Madhoun, Sulaiman K. Marafie, Dania Haddad, Motasem Melhem, Mohamed Abu-Farha, Hamad Ali, Sardar Sindhu, Maher Atari, Fahd Al-Mulla

**Affiliations:** 1Department of Animal and Imaging Core Facilities, Dasman Diabetes Institute, Dasman 15462, Kuwait; Sardar.Sindhu@dasmaninstitute.org; 2Department of Genetics and Bioinformatics, Dasman Diabetes Institute, Dasman 15462, Kuwait; dania.haddad@dasmaninstitute.org (D.H.); motasem.melhem@dasmaninstitute.org (M.M.); hamad.ali@dasmaninstitute.org (H.A.); fahd.almulla@dasmaninstitute.org (F.A.-M.); 3Department of Biochemistry and Molecular Biology, Dasman Diabetes Institute, Dasman 15462, Kuwait; sulaiman.marafie@dasmaninstitute.org (S.K.M.); mohamed.abufarha@dasmaninstitute.org (M.A.-F.); 4Department of Medical Laboratory Sciences, Faculty of Allied Health Sciences, Health Sciences Center (HSC), Kuwait University, Jabriya 046302, Kuwait; 5Medical-Surgical Pathology Department, Regenerative Medicine Research Institute, Universitat Internacional de Catalunya, 08195 Barcelona, Spain; matari@biointelligentsl.com

**Keywords:** adult skin fibroblasts, mass spectrometry, neonate foreskin fibroblasts, proteomic analysis, Wharton’s jelly-derived mesenchymal stem cells

## Abstract

Wharton’s jelly-derived mesenchymal stem cells (WJ-MSCs) are a valuable tool in stem cell research due to their high proliferation rate, multi-lineage differentiation potential, and immunotolerance properties. However, fibroblast impurity during WJ-MSCs isolation is unavoidable because of morphological similarities and shared surface markers. Here, a proteomic approach was employed to identify specific proteins differentially expressed by WJ-MSCs in comparison to those by neonatal foreskin and adult skin fibroblasts (NFFs and ASFs, respectively). Mass spectrometry analysis identified 454 proteins with a transmembrane domain. These proteins were then compared across the different cell-lines and categorized based on their cellular localizations, biological processes, and molecular functions. The expression patterns of a selected set of proteins were further confirmed by quantitative reverse transcription polymerase chain reaction (qRT-PCR), Western blotting, and immunofluorescence assays. As anticipated, most of the studied proteins had common expression patterns. However, EphA2, SLC25A4, and SOD2 were predominantly expressed by WJ-MSCs, while CDH2 and Talin2 were specific to NFFs and ASFs, respectively. Here, EphA2 was established as a potential surface-specific marker to distinguish WJ-MSCs from fibroblasts and for prospective use to prepare pure primary cultures of WJ-MSCs. Additionally, CDH2 could be used for a negative-selection isolation/depletion method to remove neonatal fibroblasts contaminating preparations of WJ-MSCs.

## 1. Introduction

Mesenchymal stem cells (MSCs) are a population of non-hematopoietic stem cells with multipotent properties that are not associated with teratoma formation [1,2]. Because of these properties, MSCs are an attractive alternative to embryonic stem cells for research and prospective clinical applications. The therapeutic potential of MSCs is not limited to their capacity to replace injured tissue cells. They also have a paracrine effect on the surrounding environment that modulates inflammation, reduces stress-induced apoptosis, and enhances revascularization [3]. MSCs have been isolated from various tissues of the human body [4,5,6], the bone marrow (BM) and adipose tissue (AT) are the main source for prospective clinical applications [7].

There are a number of limitations to the use of adult MSCs. For example, the procedure for the collection of BM-derived MSCs (BM-MSCs), which account for a small fraction of nucleated BM cells, is particularly invasive and restricted to the availability of suitable donors [8]. In addition, BM-MSCs have limited long-term proliferation and the differentiation potential is linked to the donor’s age [9]. On the other hand, AT-derived MSCs (AT-MSCs) are more abundant and the isolation procedure is less invasive. However, the expansion and differentiation of AT-MSCs are dependent on the age and health status of the donor [10]. Wharton’s jelly-derived MSCs (WJ-MSCs) continue to gain the interest of researchers as a promising alternative source of multipotent cells that do not require an invasive isolation procedure [11]. This unique population of cells is embedded within the gelatinous material of the umbilical cord, known as Wharton’s jelly [12,13]. Resembling adult MSCs, WJ-MSCs have the capacity of self-renewal and immuno-modular properties [1,14,15,16]. As multipotent cells, WJ-MSCs can be differentiated in vitro into a wide spectrum of cell types from the three germ layers or at least in part, express specific markers [17,18,19,20]. Indeed, current data are conflicting in regard to the ability of MSCs to generate terminal and functional cells from either germ layer due to the use of various isolation, proliferation, and differentiation protocols that are biased due to differences in the stimulus used to induce cell signaling [21,22,23,24] in addition to the existence of cell populations at different developmental stages.

Like MSCs, WJ-MSCs are not associated with teratoma formation upon transplantation, thus clinical applications of these cells are ethically accepted [25,26]. In fact, BM-MSCs and WJ-MSCs are believed to have a common ancestor. Nevertheless, we and others have shown differences in the differentiation potential as well as the transcriptomic and proteomic profiles [1,27,28].

MSCs are often employed in the field of regenerative medicine due to their immunomodulatory effects, which virtually erases the risk of immunorejection and eliminates the need for immunosuppressive therapy prior to transplantation [29,30]. The immunomodulatory capacity of MSCs is mediated through a cell contact-dependent mechanism and the secretion of paracrine factors [31,32,33]. Thus, crosstalk between MSCs and immune cells is sufficient to generate a homeostatic mechanism by which MSCs regulate the immune response. Although, MSCs from different sources have comparable influences on the immunophenotype [34,35], some minor differences exist. Relative to BM-MSCs, AT-MSCs reportedly have greater immunomodulatory potency due to high levels of cytokine secretion [36] and prevention of immunogenicity [35]. On the other hand, a recent report indicated that BM-MSCs possess higher immunomodulatory activities and secrete lower amounts of paracrine signaling molecules relative to AT-MSCs and WJ-MSCs [37]. These discrepancies have been attributed to differences in cultural conditions, variations in experimental setups, and, most importantly, the crosstalk between each type of MSCs and the targeted immune cell population [38]. Although further in-depth studies are needed to clarify the immunomodulatory effects of MSCs from different sources, the advantages of perinatal vs. adult MECs include the short prenatal lifespan, limited exposure to pro-aging factors, and less cellular/genetic damage that might affect cellular plasticity [39,40].

Fibroblasts, which are the most common somatic cell type, form structural frameworks and produce an extracellular matrix that supports the surrounding tissues. Fibroblasts are not terminally differentiated cells, but rather respond to stimuli that activate proliferation and differentiation potential, and also play important roles in wound healing, inflammation, angiogenesis, and tissue fibrosis [41]. Current methods for the isolation of MSC and WJ-MSC do not prevent fibroblast contamination, as these cell populations share a spindle-like morphology, expression of common surface antigens, and plastic adherence properties [42]. In addition to reducing the yield of these multipotent cell populations, fibroblast contamination may increase the risk of damage to MSCs and WJ-MSCs, resulting in senescence or cell death, reduced differentiation potential, or even tumorigenic transformation following transplantation [2,42].

Currently, there is no consensus on a single surface marker to differentiate WJ-MSCs from fibroblasts of various sources, which is essential for the isolation of pure and authentic populations of WJ-MSCs that can be introduced into damaged tissues or organs without passaging in tissue culture, for prospective clinical applications. In this study, the proteomic profiles of membrane-bound proteins extracted from WJ-MSCs, neonatal foreskin fibroblasts (NFFs), and adult skin fibroblasts (ASFs) were characterized using nanoscale liquid chromatography coupled to tandem mass spectrometry (Nano LC-MS/MS). Gene expression analysis at the transcriptional and protein levels indicated that ephrin type-A receptor 2 (EphA2) is a candidate surface-specific protein for the identification of WJ-MSCs, whereas cluster of differentiation (CD)H2 and Talin2 are markers for NFFs and ASFs, respectively.

## 2. Results

### 2.1. Analysis of Differentially Expressed Proteins Detected by Nano LC-MS/MS 

To illustrate differences in the proteomic patterns between different passages of WJ-MSCs, ASFs, and NFFs, membrane-fraction protein extracts were digested, and the generated peptides were detected by Nano LC-MS/MS. Although there were some variations in the number of proteins identified between passages, most proteins were common among different passages of the same cell type. In total, 958, 866, and 813 proteins were shared among the different passages of WJ-MSCs, ASFs, and NFFs, respectively (Figure 1A). Then, we compared the number of proteins shared among different cell-types and identified 905 proteins that were commonly expressed among WJ-MSCs, ASFs, and NFFs (Figure 1B). Among the cell types, a total of 97 differentially expressed proteins were identified, including 56 that were unique to WJ-MSCs, 23 unique to NFFs, and 18 unique to ASFs. Moreover, 41 proteins were shared between WJ-MSCs and NFFs, 60 between WJ-MSCs and ASFs, and 23 between ASFs and NFFs.

The identified proteins were first screened to identify all membrane-bound proteins that could be potential candidate cell surface markers. Of the initial 1126 screened proteins, 454 were found to have a transmembrane domain, and then categorized according to involvement in biological processes, molecular functions, and cellular localization using gene ontology (GO) enrichment methods (Figure 2). Of the proteins involved in biological processes, most had intracellular transport functions or targeted the cellular membranes and organelles. Several proteins were associated with RNA processing, cell biogenesis, and organization (Figure 2A). Of the proteins involved in molecular functions, most were either associated with RNA binding functions, cell signaling, or the ribosomal complex (Figure 2B). Most of the detected proteins were present in vesicles, membrane-bound organelles, or secretory exosomes (Figure 2C).

Quantitative MS was performed to identify proteins with differential expression patterns in different cell types. These proteins were compared and classified based on their cellular localization, as well as involvement in biological processes and molecular functions (Table 1 and Table 2). Many of the identified proteins were common among different cell types, while others were specific to a particular cell type. For example, proteins involved in plasma membrane rafts and cell–cell junctions were specific to a particular cell type, including EphA2 in WJ-MSCs, the cytoskeletal anchoring protein Talin2 (TLN2) in ASFs, and neuronal (N)-cadherin (CDH2/CD325) in NFFs (Table 1, Cytoplasmic membrane). In the mitochondria, the adenosine di/triphosphate (ADP/ATP) translocase 1/solute carrier family 25 member 4 (SLC25A4) and voltage-dependent anion-selective channel protein VDAC3 appear to be specific to WJ-MSCs, while mitochondrial superoxide dismutase (SOD2) was identified in both WJ-MSCs and ASFs. The integrin alpha subunit ITGA2 (CD49b) was common to both WJ-MSCs and NFFs, while lipase maturation factor (LMF2) was detected specific to fibroblasts (both ASFs and NFFs).

In addition, the dataset contains shared and cell type-specific proteins within many functional classes, thereby revealing important differences in the protein profiles of specific cell types (Table 2). For example, signaling proteins known to be involved in the tricarboxylic acid cycle were detected in all of the cell types studied. On the other hand, the notch signaling protein ANXA4 was specific to fibroblasts. The Wnt signaling molecules CTHRC1 and CDH2 were only detected in WJ-MSCs and NFFs respectively, whereas Ras homolog family member A (RHOA) and ubiquitin A-52 residue ribosomal protein fusion product 1 (UBA52) were present in both WJ-MSCs and ASFs. Similarly, the studied cell types had notable differentially expressed proteins involved in other biological mechanisms, including metabolic and oxidation-reduction, cell adherence, cell component transportation, and biogenesis (Table 2). Thus, the proteomic dataset provides an important resource of cell-surface proteins present on WJ-MSCs that could be used in future functional studies.

### 2.2. Quantitative RT-PCR Analysis of Gene Products (Proteins) Identified by Mass spectrometry (MS) Screening

While generally a good indicator of protein translation in the cells, the mRNA level is not always correlated with the presence of the encoded protein [43]. Thus, to verify whether the relative expression levels of the proteins detected via MS in different cell types could have been predicted by the mRNA levels; quantitative reverse transcription polymerase chain reaction (qRT-PCR) analysis was performed to analyze the gene expression (Figure 3). The expression pattern of CD73, a well-known marker of WJ-MSCs was also analyzed, which showed comparable expression levels in both WJ-MSCs and fibroblasts (Figure 3).

The mRNA expression profiles of the membrane-bound proteins EphA2, SLC25A4, TLN2, LMF2, and CD49b were similar by qRT-PCR analysis and the quantitative MS spectra. EphA2 and SLC25A4 were predominantly expressed by WJ-MSCs. The expression levels of AphA2 in ASFs and NFFs were only 13% and 27% relative to those of WJ-MSCs, respectively. While SLC25A4 transcripts in both fibroblast cell types were <10% relative to those of WJ-MSCs. On the other hand, TLN2 and LMF2 genes were notably expressed by fibroblasts. In both ASFs and NFFs, the expression levels of TLN2 and LMF2 were 43- and 12-fold, and 17- and 15-fold relative to that of WJ-MSCs, respectively (Figure 3). CD49b was expressed mainly by WJ-MSCs, to a lesser extent by NFFs (40% to that of WJ-MSCs), and as low as 15% in ASFs. Alternatively, mRNA levels of VDAC3, SOD2, and plectin-1 (PLEC1) did not reflect the associated protein expression levels determined by MS analysis (Table 3), whereas SOD2 expression levels were 20-fold higher in WJ-MSCs than fibroblasts. The mRNA expression levels of VDAC3 and PLEC1 were equivalent in all cell types (Figure 3).

The expression levels of the integrin subunit integrin alpha-5 (ITGA5, CD49e), insulin-like growth factor 2 mRNA-binding protein 3 (IGF2BP3), the RNA transcription, translation and transport factor CLE7, the vascular-endothelial VE-cadherins (CDH5, CD144), and the actin-binding protein Nexilin were also analyzed. The mRNA quantifications by qRT-PCR results revealed that CD49e, CLE7, CDH5, and Nexilin were primarily expressed by WJ-MSCs. In fibroblasts, CD49e transcripts were 50% lower than in WJ-MSCs, while CLE7 transcripts were absent in NFFs and accounted for less than 24% in ASFs relative to WJ-MSCs (Figure 3). CDH5 and Nexilin mRNAs were >4- and >6-fold higher than in WJ-MSCs fibroblasts, respectively (Figure 3). On the other hand, IGF2BP3 was expressed mostly in WJ-MSCs and ASFs, while CDH2 transcripts were predominately expressed by NFFs (>80-fold vs. other cell-types, Figure 3).

### 2.3. Western Blot Analysis

Western blot analysis of selected proteins was performed to validate the quantitative proteomic results obtained by MS and to assess correlations with the mRNA levels determined by qRT-PCR (Figure 4). As compared to the findings of MS and qRT-PCR, the protein levels of CD49b, as determined by Western blot analysis, were approximately 8-fold higher in WJ-MSCs and NFFs relative to ASFs. On the other hand, levels of mitochondrial SOD2 were higher in WJ-MSCs, in agreement with their mRNA levels, but were not detected in ASFs, as predicted by the proteomics approach (Table 3). Western blot analysis using specific antibodies barely detected SOD2 proteins in fibroblasts (Figure 4). Protein levels of PLEC1, Nexilin, TLN2, and CD49e replicated the expression patterns determined by qRT-PCR analysis. However, the expression levels of these proteins in different cell types did not mimic those predicted by the proteomics approach. PLEC1 and Nexilin proteins were comparably expressed by all cell types, whereas CD49e proteins were equivalently detected in WJ-MSCs and ASFs but were 50% lower in NFFs. On the other hand, TLN2 was predominately observed in ASFs and seldomly detected in WH-MSCs or NFFs. Protein levels of LMF2 were 5-fold greater in ASFs relative to WJ-MSCs, but not expressed in NFFs, as observed by both qRT-PCR analysis and the initial MS screening. Similarly, EphA2 and SLC25A4 were predominantly expressed by WJ-MSCs, as predicted by the proteomics approach and validated by qRT-PCR (accounting for <20% in fibroblasts, Figure 4 and Table 3).

### 2.4. Fluorescence Microscopy

Next, fluorescence microscopy was used to visualize the expression patterns and the cellular localizations of proteins of interest in WJ-MSCs, ASFs, and NFFs (Figure 5). SOD2 was detected predominantly in WJ-MSCs with lower expression seen in NFFs. CD49b was mostly expressed by WJ-MSCs, with some expression detected in ASFs. On the contrary, TLN2 was observed only in ASFs, confirming the results obtained by Western blot analysis. LMF2 was expressed mainly in ASFs, with some minor expression patterns in NFFs, correlating with the pattern detected by Western blot analysis. The myosin light chain kinase MLCK1 was expressed mostly in NFFs, with lower levels in ASFs, similar to the expression profile of N-cadherin (CD325).

## 3. Discussion

Due to the potential use in regenerative medicine, WJ-MSCs have continued to attract the interest of academic and medical communities in recent years. Despite a multitude of studies, the isolation of WJ-MSCs from contaminating cell populations remains difficult, particularly with fibroblasts. In order to develop more convenient and targeted methods of purification, cell-specific surface markers were identified to precisely isolate WJ-MSCs.

According to the current criteria defined by the International Society for Cellular Therapy, MSCs express the membrane proteins CD105, CD73, and CD90, but not CD45, CD34, CD14, or CD11b, CD79α, or CD19, and HLA-DR [11]. Besides the present study, several previous studies have reported that many of these surface markers are shared with fibroblasts [2,11,42,44,45,46,47]. On the other hand, MSCs express various fibroblast proteins, such as collagen, vimentin, fibroblast surface protein, heat shock protein 47, and α-smooth muscle actin [2,44]. Thus, MSCs are a heterogeneous cell population that lacks a specific surface biomarker, thereby rendering identification and characterization of MSCs rather challenging. In this study, 454 membrane-bound proteins differentially expressed by these cell types were identified. Here, a few highly expressed markers were selected for further analysis, which included EphaA2, CDH5, and the integrin alpha subunits CD49b and CD49e.

EphA2 belongs to the Ephrin receptor subfamily of the protein-tyrosine kinase family. EphA2 and its ligand Ephrin play important roles in cellular migration, survival, and differentiation [48]. In general, Ephrin receptors mediate cell-to-cell binding, leading to contact-dependent bidirectional signaling to neighboring cells [49]. During embryogenesis, Ephrin receptors mediate neuron differentiation, neural-tube formation, and development of the early hindbrain [50]. Ephrin receptors influence a niche of stem cells. In the present study, EphA2 was primarily expressed by WJ-MSCs supporting its role in cell self-renewal and differentiation [51]: the two major characteristics of stem cells that fibroblasts lack. The results of previous gene expression studies indicate that MSCs derived from BM and AT express a wide range of Ephrin receptors, including EphA2 [48], whereas MSCs isolated from the umbilical cord blood expresses EphB2 [52]. In support of our data, proteomic analysis identified EphA2 as a marker of MSCs derived from the placenta, a cell type that is developmentally related to WJ-MSCs [53] and is believed to secrete prostaglandin E2, an anti-fibrosis and immunomodulator marker [54]. Together, these data indicate that EphA2 is an important surface marker of WJ-MSCs.

CDH5 or VE-cadherin have been previously described to be a surface marker for Adult cardiac progenitor/stem cells but not BM-MSCs [55] or WJ-MSC [56], here we found it to be a good surface marker for WJ-MSCs and to a lesser extent to NFFs. CDH5 is a calcium-dependent cell adhesion protein that ensure integrity of blood and lymph vessels and play an important role in vasculogenesis, angiogenesis, vessel leakage, and leukocyte trafficking [57]. Inhibition of CDH5 expression, by miRNA-6086, blocks human embryonic stem cells’ differentiation into endothelial cells [58]. Together, the elevated levels of CDH5 in WJ-MSCs supports their prospective differentiation into endothelial cells or possible other lineages.

Integrins play an essential role in cellular adhesion and cell surface-mediated signaling. CD49b is integrin alpha 2 subunit, which, in combination with integrin beta 1 subunit, forms a receptor for collagen, collagen C-propeptides, fibronectin, laminin, and E-cadherin [59]. Ligand recognition and binding occur mainly through the alpha subunit of the integrin heterodimer. Once activated, the receptor initiates downstream signaling events engendering changes in cell migration, survival, and growth [60]. Interestingly, integrin activation via intracellular ligands has also been reported. The binding of Talin 2 to the integrin beta subunit leads to a conformational change to its transmembrane domain, leading to integrin activation [61].

Albeit BM-MSCs and fibroblasts express similar levels of CD49b [62]; in the present study, mRNA and protein expression levels of CD49b were 2.5-fold higher in WJ-MSCs than in NFFs. This pattern is corroborated by two large transcriptomic and proteomic studies [63,64], which were generated the Human Protein Atlas (HPA). In the HPA, CD49b is mostly expressed by endothelial cells derived from the umbilical vein and to a lower extent in NFFs. The mRNA and protein expression patterns of CD49e observed in this study also mimicked those from the HPA database: mostly expressed by endothelial cells from the umbilical vein and lower expression in NFFs [59].

While several studies have screened and identified surface markers of MSCs, the identification of MSC-specific surface markers remains challenging. A systematic review of available information noted a great discrepancy in the expression patterns of several surface markers of MSCs in different studies [65]. A possible explanation for this discrepancy is related to the origin or heterogeneity of the MSCs used in different studies. Alternatively, these differences could possibly be related to the different proliferative stages of the cells in culture. In any case, further studies are needed to validate our preliminary findings as well as extrapolate the findings of previous studies to overcome inconsistencies regarding cell surface marker profiles of MSCs with the potential advantage of culture purification of WJ-MSCs via negative or positive selection. A limitation to this study is the comparison of data with only two fibroblast cell lines. Hence, the results must be interpreted with caution.

To the best of our knowledge, this is the first study that aimed to identify specific cell markers for WJ-MSCs that are not present in fibroblasts of neonate or adult origin. We confirmed EphA2 as a potential cell surface marker that distinguishes WJ-MSCs from fibroblasts and can be used to prepare pure WJ-MSCs primary cell cultures. CDH5 or VE-Cadherin can be also used as a surface marker, however it would yield a less pure WJ-MSCs population. Whereas, a negative-selection isolation process can be devised using CDH2 to remove neonatal fibroblasts, commonly encountered in WJ-MSCs preparations. Currently, we are aiming to use these surface-specific markers to prepare pure cell cultures from a wide range of MSCs derived from several tissues, including bone marrow, adipose tissue, and dental pulp.

## 4. Materials and Methods

### 4.1. Ethical Permission and Procurement of Human Samples

The study protocol approved by the Ethical Review Committee of the Dasman Diabetes Institute (protocol No: 2013-009) and conducted in accordance with the World Medical Association Declaration of Helsinki—Ethical Principles for Medical Research Involving Human Subjects. Human WJ-MSCs, NFFs, and ASFs were purchased from the ATCC (PCS-500-010, CRL-2522, and PCS-201-012, respectively).

### 4.2. Culture and Maintenance of WJ-MSCs, NFFs, and ASFs 

WJ-MSCs were cultured in Dulbecco’s modified Eagle’s medium (DMEM)/Hams’ F12 medium (1:1 v/v; Invitrogen Corporation, Carlsbad, CA, USA) supplemented with 15% MSC-qualified fetal bovine serum (FBS; Invitrogen Corporation) and 100 U/mL of penicillin-streptomycin (Gibco, Carlsbad, CA, USA). WJ-MSCs (passage 2) were cultured at approximate densities of 300–1000 cells/cm^2^ in 100 mm plates to generate colony-forming units. After 7–10 days, the culture medium was removed, cells were washed with 1× phosphate buffer saline (PBS), and cell colonies were transferred into individual 100 mm plates for expansion. NFFs and ASFs were cultured in DMEM supplemented with 10% FBS and 100 U/mL of penicillin-streptomycin. At 60% confluency, cell cultures were harvested for passaging and/or scraped for protein isolation, RNA extraction, or immunofluorescence analysis.

### 4.3. Preparation of Protein Extract for MS Analysis

Three replicates from three different passages (between passage 2 and 5) for each cell type (WJ-MSCs, NFFs, or ASFs) were used for the proteomic analysis. The cells were washed with cold PBS, lysed in 500 µL of lysis buffer (6 M urea, 4% CHAPS hydrate) supplemented with a mini complete protease inhibitor cocktail (Roche Diagnostics, Laval, QC, Canada) for 30 min at 4 °C, and then homogenized. Following centrifugation at 14,000 rpm for 30 min at 4 °C, protein concentrations were determined by Bradford method using γ-globulin as a standard, and proteins were subjected to centrifugal proteomic reactor as described previously [66]. Briefly, 20 µg of total proteins were mixed with 10 µL of a strong cationic exchange BcMag™ SCX Magnetic bead slurry (Bioclone, San Diego, CA, USA) and vigorously vortexed in 1 mL of 5% formic acid. The mixture was centrifuged, and the pellet was resuspended in 1 mL of 0.5% formic acid. Then, 20 µL of a reducing solution (150 mM ammonium bicarbonate, 20 mM dithiothreitol) was added to the samples for 15 min at 56 °C with constant rotation. Then, 20 µL of an alkylating solution (150 mM ammonium bicarbonate, 100 mM iodoacetamide) was added for 15 min at room temperature. The alkylation reaction was stopped by the addition of 1 mL of 0.5% formic acid. The beads were centrifuged, and the proteins were digested with trypsin at 37 °C for 24 h with constant shaking. The generated peptides were eluted with 1 mL of 5% formic acid, lyophilized to complete dryness in a speed-vac, and stored at −20 °C until subjected to MS analysis.

### 4.4. Nano Liquid Chromatography with Tandem Mass Spectrometry (LC-MS/MS) Analysis

Proteomics were analyzed using LC-MS/MS (LTQ-Orbitrap Velos; Thermo Fisher Scientific, Waltham, MA, USA) as described previously [67]. Briefly, peptides were resuspended in 5% formic acid and then loaded on a C18-A1 easy column for desalting (Proxeon Biosystems A/S, Odense, Denmark). Desalted peptides were than directed to a C18-A2 analytical easy column and eluted at a gradient of 5% to 35% acetonitrile with 0.1% formic acid for 120 min (Proxeon Biosystems A/S). Full MS scanning was performed at a resolution of 60,000 and the top 20 spectra were obtained in the data-dependent acquisition mode. Raw data files were analyzed using Maxquant 1.3.0.5 software (Thermo Fisher Scientific GmbH, Driesch, Germany) using the Sequest tandem MS data analysis program and the Mascot search engine against the *Homo sapiens* International Protein Index protein sequence database version 3.68 (European Bioinformatics Institute, Cambridge, UK), as described previously [67], with the following search parameters: the digestive enzyme trypsin, with two miss-cleavages permits, fixed modification, carbamidomethyl of cysteine, variable modification, methionine oxidation, and a false discovery rate (FDR) of 1% for peptide confidence. The identified proteins were quantified using Sieve software version 1.3 (Thermo Fisher Scientific GmbH). Ingenuity pathway analysis software (Ingenuity Systems, Inc., Redwood City, CA, USA) was used for analyses of molecular functions and protein networks.

### 4.5. Functional Enrichment Analysis

Functional enrichment analyses were performed using the Database for Annotation, Visualization and Integrated Discovery (DAVID) functional annotation tool (http://david.abcc.ncifcrf.gov/). All identified proteins from different cell types with an FDR threshold of <10% and default options and human Encyclopedia of DNA Elements (ENCODE) as a background were searched according to the following annotation categories: GeneOntology cellular components (GOTERM_CC_FAT), molecular functions (GOTERM_MF_FAT), and biological processes (GOTERM_BP_FAT), along with Protein Analysis Through Evolutionary Relationships (PANTHER) molecular functions, biological processes and pathways, Kyoto Encyclopedia of Genes and Genomes pathways, and the InterPro and Simple Modular Architecture Research Tool (SMART) protein domains (for all analyses, a FDR threshold of 10% was applied for muliple testing). Only the following GO terms were significantly enriched in our gene list (with 446/456 genes having a DAVID gene ID): “extrinsic to membrane” (GOTERM_CC_FAT), “cytoskeletal protein binding,” “purine ribonucleotide binding,” and “ribonucleotide binding” (GOTERM_MF_FAT).

### 4.6. Western Blot and Immunofluorescence Assays

Western blot analysis was performed as described previously [68]. Briefly, cells were harvested and lysed in modified radioimmunoprecipitation assay buffer. Cell lysates were quantified using the Pierce BCA Protein Assay Kit (Thermo Fisher Scientific) and equal amounts of protein were resolved on 8%–12% polyacrylamide gels and transferred to polyvinylidene fluoride membranes (EMD Millipore Corporation, Billerica, MA, USA). After blocking, the membranes were blotted with the corresponding primary and horseradish peroxidase-linked secondary antibodies (Appendix A). Proteins were visualized using Amersham ECL Prime Western Blotting Detection Reagent (GE Healthcare Life Sciences, Chicago, IL, USA). Images were captured using the VersaDoc™ MP 5000 imaging system (Bio-Rad Laboratories, Hercules, CA, USA).

Immunofluorescence was performed as described previously [69]. Briefly, cells were cultured overnight on glass coverslips coated with 0.1% gelatin, fixed with 4% paraformaldehyde for 15 min, permeabilized with 10% Triton X-100 for 30 min, and incubated overnight with appropriate antibodies (Appendix A). Primary antibodies were conjugated with Alexa Fluor 594 or Alexa Flour 488 using APEX Antibody Labeling Kits (Invitrogen Corporation). Fluorescent images were captured using a confocal laser-scanning microscope (LSM 710; Carl Zeiss AG, Oberkochen, Germenty) as previously described [70].

### 4.7. RNA Extraction, cDNA Synthesis, and qRT-PCR Reactions

Total RNA was extracted from cells using the total RNA purification kit (Norgen Biotek Corp, Thorold, ON, Canada) in accordance with the manufacturer’s protocol. Total RNA was quantified using a NanoDrop 2000c spectrophotometer (Thermo Fisher Scientific) and RNA integrity was evaluated by 2% agarose gel electrophoresis (data not shown). First-strand cDNA was synthesized from 200 ng RNA by reverse transcription using the QuantiTect Reverse Transcription Kit (Qiagen Inc., Valencia, USA). The qRT-PCR reactions were performed as described elsewhere [71]. Primer pairs (Appendix A) were selected from the PrimerBank database (https://pga.mgh.harvard.edu/primerbank/) [72] and tested for equivalent efficiency. Quantitative polymerase chain reaction (qPCR) was performed on the ABI7900 system (Applied Biosystems, Waltham, MA, USA) using SDS software. Relative gene expression was calculated using the comparative cycle threshold (Ct) value method as previously described [73]. Results were normalized to Glyceraldehyde 3-phosphate dehydrogenase (GAPDH) and expressed relative to WJ-MSCs’ gene expression.

### 4.8. Statistical Analyses

All experiments and assays were done in technical duplicates or triplicates for three biological samples. Results were compounded and statistical significance was estimated with a two-tailed Student’s *t*-test assuming equal variance performed in GraphPad Prism version 8.0. Data were presented as mean ± standard error of the mean (SEM) as previously described [73].

## Figures and Tables

**Figure 1 ijms-21-06437-f001:**
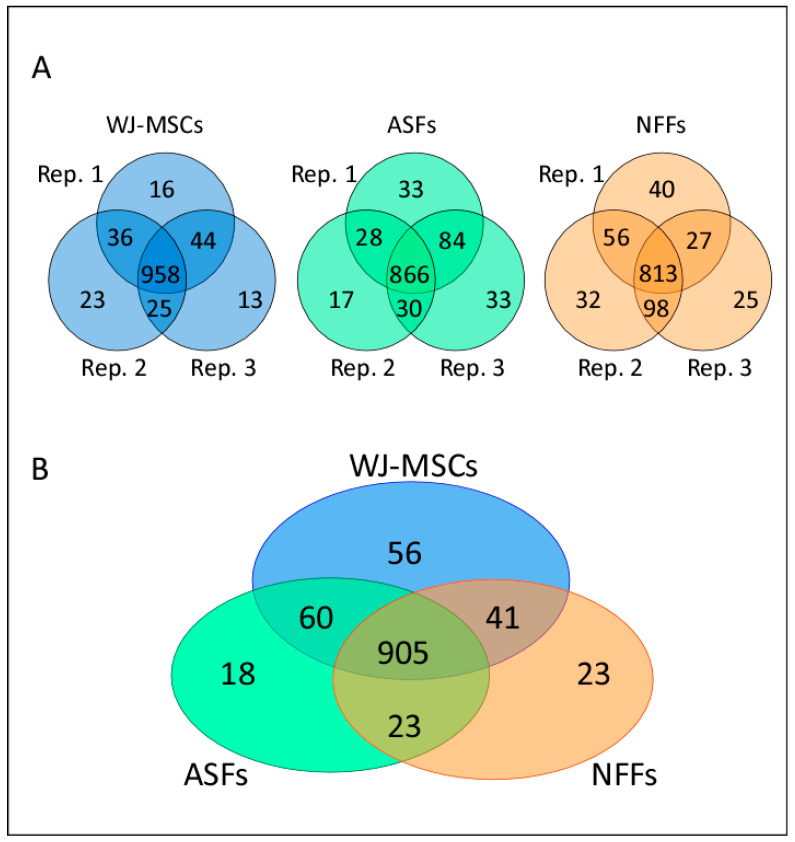
Proteomic analysis of WJ-MSCs, ASFs and NFFs. (**A**) A Venn diagram illustrating the number of proteins expressed by three different cell passages of WJ-MSCs, ASFs, and NFFs. For each cell type, the overlapping areas between the cell passage populations indicate the numbers of commonly expressed proteins. (**B**) A Venn diagram showing the number of proteins expressed by each cell type. The overlapping areas indicate the numbers of proteins commonly expressed by WJ-MSCs, ASFs, and NFFs, as identified by MS analysis. Rep, replicate.

**Figure 2 ijms-21-06437-f002:**
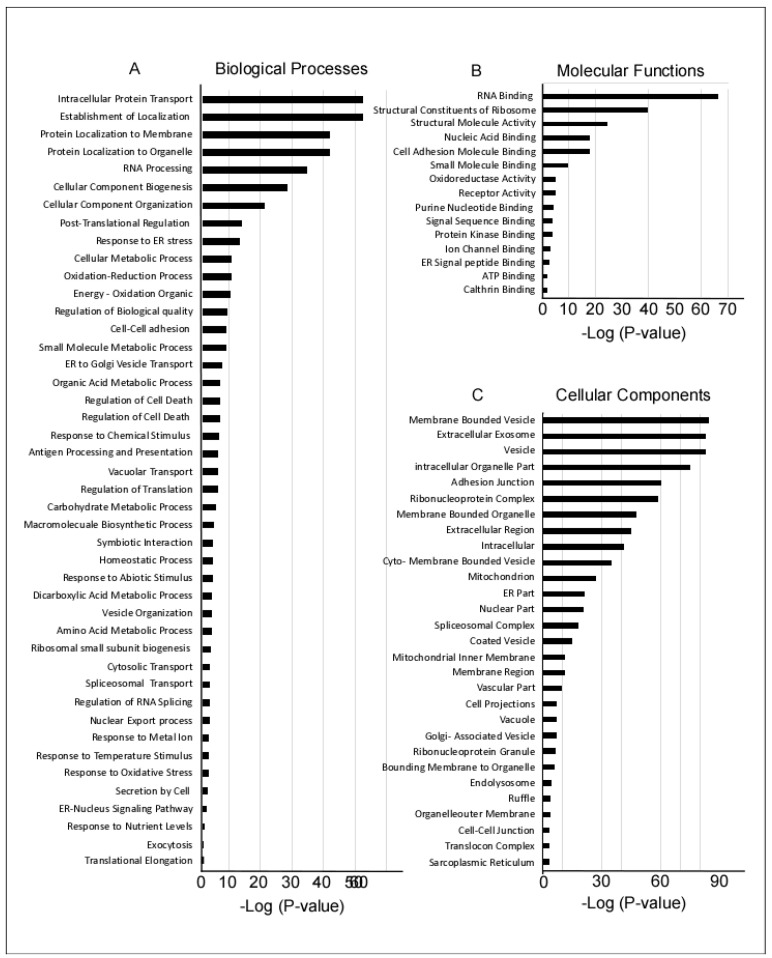
Pathways and gene ontology (GO) enrichment analysis of 1126 proteins expressed by WJ-MSCs, ASFs, and NFFs. The GO annotation enrichment score [−Log2 (*p*-value)] analysis of proteins involved in biological processes (**A**), molecular functions (**B**), and cellular components (**C**), as identified by MS and in proteomic analysis.

**Figure 3 ijms-21-06437-f003:**
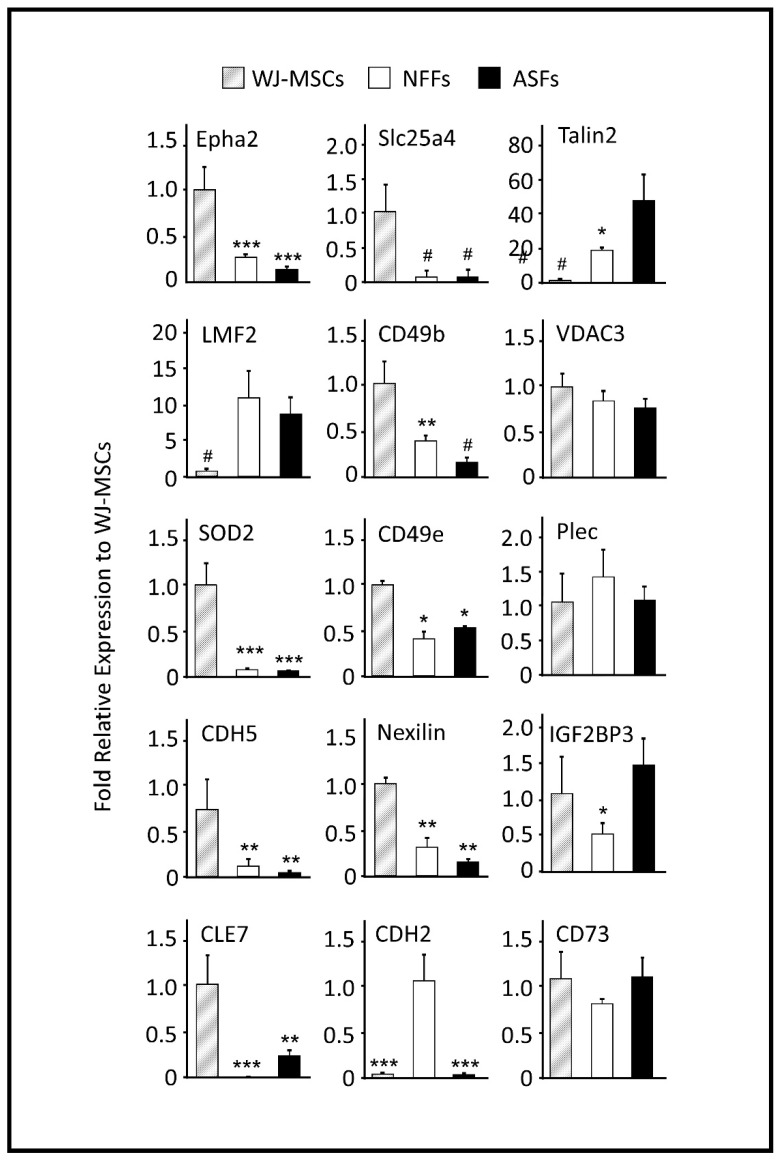
qRT-PCR analysis of 15 selected genes differentially expressed by WJ-MSCs, ASFs, and NFFs. Relative quantification was calculated by comparing gene expression levels in ASFs and NFFs with corresponding WJ-MSCs expression, which was set to 1 (control sample). Gene expression was initially normalized to the geometric mean for the housekeeping genes β-actin, Glyceraldehyde 3-phosphate dehydrogenase (GAPDH), and S18, as references. Data are presented as the means ± standard deviation of six qRT-PCR assays (technical duplicate of three biological samples). * *p* < 0.05, ** *p* < 0.01, and *** *p* < 0.001 are significant to the higher bar. # *p* < 0.01 is significant to all other bars.

**Figure 4 ijms-21-06437-f004:**
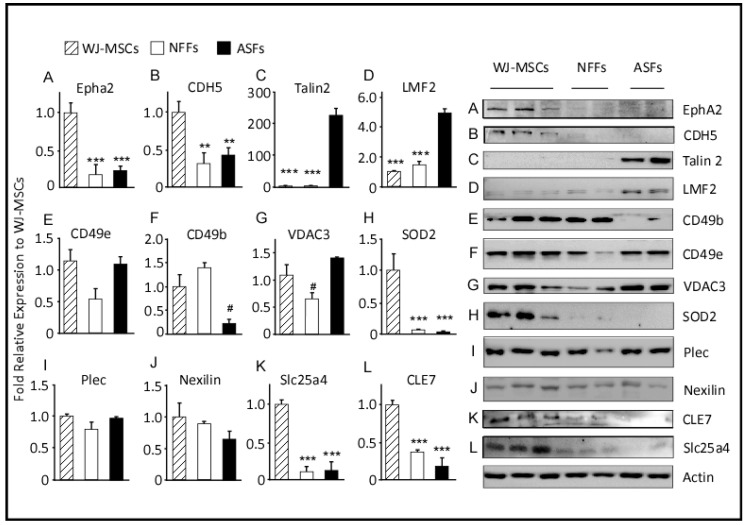
Expression levels of proteins differentially expressed by WJ-MSCs, ASFs, and NFFs. Representative Western blots of 12 expressed proteins. Normalized proteins are expressed relative to their prospective expression in WJ-MSCs. Data are presented as the means ± standard error of the mean of three independent assays. ** *p* < 0.01, and *** *p* < 0.001 are significant to the higher bar. # *p* < 0.01 is significant to all other bars.

**Figure 5 ijms-21-06437-f005:**
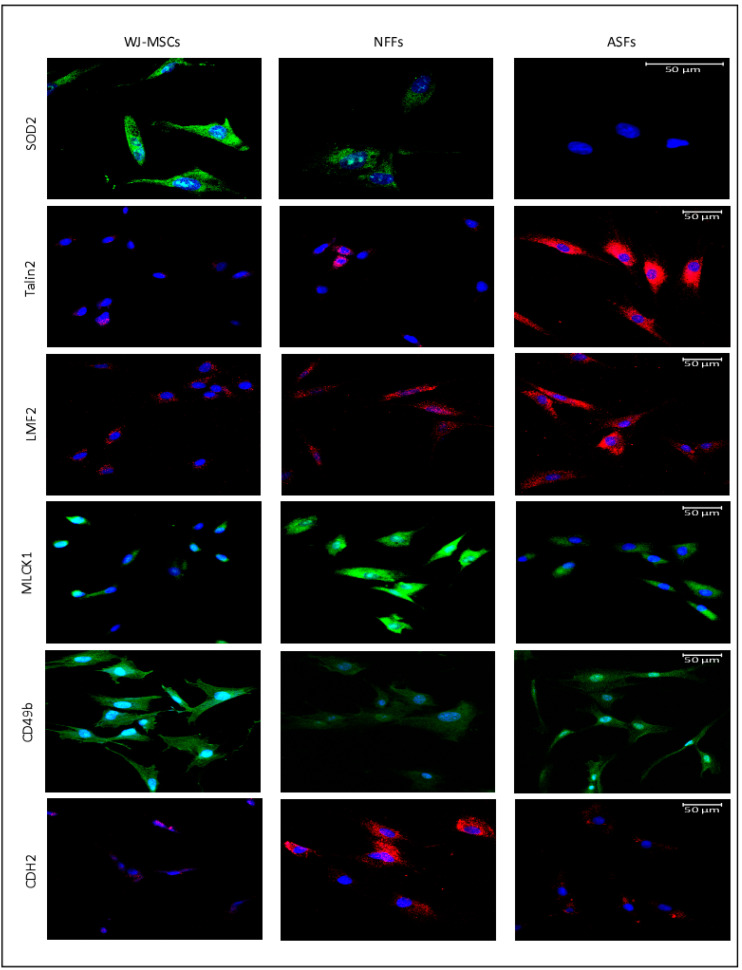
Immunofluorescence of selected proteins expressed in WJ-MSCs, NFFs, and ASFs. Representative confocal laser microscopy images of immunofluorescence for primary cells using APEX antibody labeling system for conjugating the indicated primary antibodies as described in the Materials and Methods Section. Shown at 400× magnification. Nuclei were stained with 4′,6-diamidino-2-phenylindole (DAPI).

**Table 1 ijms-21-06437-t001:** Cellular localization of the membrane-bound proteins differentially expressed by WJ-MSCs, ASFs, and NFFs. − means un-detected.

Cellular Localization	Unique to WJ-MSCs	Unique to NFFs	Unique to ASFs	WJ-MSCs and NFFs	WJ-MSCs and ASFs	ASFs and NFFs
Extracellular Exosome	ANP32B, ALDH1A3, ATP1B3, ATP2B1, DDX19B, DYNC2H1, EIF3E, GRHPR, HSPA2, ICAM1, NUCB2, PFKL, PFDN2, PDCD5, PCMT1, RAB5A, TMEM106B, TUBB3, VDAC3	CDH2, EFHD1, RAB3B, RAB3B, SNRPE, TPP1	AK1, ATP6V1A, GNAL, NDRG1, QDPR, SIRPA, TUBB4A, UBE2V2	COLEC12, DSTN, DLST, ECHS1, IGF2R, IGFBP7, IDH1, IDH2, NDUFB4, PRDX3, PLD3, REEP5, SCAMP3, STX12, TMED9, TPM3, USP14	ADH5, CAPN1, CAPN2, DNAJB4, EMILIN1, GLIPR2, GFPT1, ITSN2, LTA4H, HLA-B, HLA-C, MARS, PDLIM2, PEPD, PSMC6, PSMB6, RHOA, RHOC, RAC1, RAC2, RAC3, RNH1, RPS17, STK10, SOD2, TXNL1, TIMP3, FLJ44635, TCEB2, TPT1, UBE2L3	ANXA4, CYB5A, GNB2, GNB4, LAMTOR1, PCK2, PTGS1, PRKCA, RPLP1, SNX18
Cell–Cell Adhesion	DDX6, EPHA2, GCN1, LIMS1, BZW2, PCMT1	CDH2, GOLGA3, STAT1, TMOD3	TLN2	IDH1, KTN1, MACF1	EFHD2, SWAP70, MYH9, PSMB6, RHOA, TJP2, TWF1	OXTR, SNX1
Cellular Adherence	DDX6, ICAM1, LIMS1, TGFB1I1	STAT1	TNS3	DPP4, IGF2R), ITGA2, IDH1	CAPN2, RAC1, TLN2	CAPN1
Mitochondria	C1QBP, LETM1, PYCR1, PYCR2, SLC25A4, SQRDL, VDAC3	EFHD1	–	BRI3BP, NDUFB4, DLST, ECHS1, IDH2, PRDX3, PITRM1	ALDH18A1, DLD, GLUD1, OGDH, SOD2	CYB5A, PCK2, PRKCA
Mitochondrial Envelope	LETM1, SLC25A4, SQRDL, VDAC3	EFHD1, SLC25A1	–	NDUFB4	ALDH18A1, OGDH, SOD2	CYB5A, COX1, PRKCA
Endoplasmic Reticulum (ER)	APOL2	MLEC, PML	–	PLD3, REEP5, TMED9	CAPN2, HLA-B, HLA-C, RHOA, RAC1, SEC24D	CYB5A, LMF2, PTGS1
ER Membrane	APOL2	RAB2B	–	KTN1	HLA-B, HLA-C, RAC1, SEC24D	CYB5A, PTGS1
Nuclear Parts	C1QBP, DDX6, LSM2, NUP93, TGFB1I1, WDR36	STAT1, SNRPE, SART3, TPP1	–	ADAR, DLST, ECHS1, IGF2R, IDH1, NDUFB4, PITRM1, SPARC, TP53BP1, U2AF1	GLUD1, OGDH, PPP3CA, SRRT, SF1, SOD2, TCEB2	PRKCA
Plasma Membrane Raft	EPHA2, RAB5A	CDH2	TLN2	ATP1B3, MACF1, TPM1	PRKAR1A, RAC1, TWF1	–
Cytoplasm Membrane		RAB3B, TPP1	PACS1	ATP1B3, DPP4, IGF2R, SPARC, STX12, USP14	HLA-B, HLA-C, RAC1, SEC24D	AP1B1
Intracellular Membranes	APOL2, C1QBP, DDX19A, DDX19B, DDX6, LETM1, LIMS1, LSM2, NUCB2, NUP93, PFKL, SLC25A4, SQRDL, TGFB1I1, VDAC3	EFHD1, RAB2B, RAB3B, STAT1, SNRPE, SLC25A1, SART3, TPP1	PACS1, RANBP2, RGPD3, RGPD4, RGPD5, RGPD6, RGPD8, TNS3	ADAR, ATP1B3, DLST, DPP4, ECHS1, IGF2R, ITGA2, IDH1, KTN1, NDUFB4, PITRM1, SCRN1, SPARC, STX12, TPM2, TP53BP1, U2AF1, USP14	ALDH18A1, CAPN1, CAPN2, GLUD1, GLUD2, HLA-B, HLA-C, OGDH, PRKAR1A, PPP3CA, RAC1, SEC24D, STRAP, SRRT, SF1, SOD2, TLN2, TCEB2, TPM3	AP1B1, COX1, PGP, PTGS1, PRKCA, RPLP1, ADH5
Spliceosome	LSM2	SNRPE	–	ADAR	SNRPA, SF1	–

WJ-MSCs, Wharton’s jelly-derived mesenchymal stem cells; ASFs, Adult skin fibroblasts; NFFs, Neonate foreskin fibroblasts.

**Table 2 ijms-21-06437-t002:** Involvement of the identified membrane-bound proteins in signal transduction pathways and biological processes that were differentially by WJ-MSCs, ASFs, and NFFs. − means un-detected.

Signaling Pathways	Unique to WJ-MSCs	Unique to NFFs	Unique to ASFs	WJ-MSCs and NFFs	WJ-MSCs and ASFs	ASFs and NFFs
Signal Transduction Pathway						
Wnt	CTHRC1	CDH2	–	–	RHOA, UBA52	–
NF-κB	–	–	–	–	PSMB6	–
Notch	–	–	–	–	–	ANXA4
Interferon	–	–	–	ADAR, STAT1	HLA-B, HLA-C	–
Insulin/Glucose	NUP93, PFKL	–	ATP6V1A, RANBP2	–	–	–
Amino Acid Biosynthesis	PYCR1, PYCR2	–	–	–	ALDH18, GLUD1	–
Tricarboxylic Acid Cycle	–	–	–	DLST, IDH1, IDH2,	DLD, OGDH	–
Intracellular Transport	DDX19A, DDX19B, NUP93, RPL9	SNRPE	RANBP2, RGPD3, RGPD4, RGPD5, RGPD6, RGPD8	ADAR, STX12, U2AF1	PPP3CA, RPS17, SEC24D	AP1B1, RPLP1
RNA Processing	C1QBP, LSM2, RPL9, WDR36	SNRPE, SART3	–	ADAR, RPS17, U2AF1	STRAP, SRRT, SNRPA, SF1	RPLP1
Cell Component Biogenesis	DDX6, LIMS1, WDR36, C1QBP, ICAM1, NUP93, PFKL, VDAC3	SNRPE, SART3	–	NDUFB4, ADAR, ITGA2, STX12	SEC24D, PRKAR1A, RAC1, STRAP, SF1, SOD2, TLN2, TCEB2	PRKCA, RPLP1
Oxidation-reduction Process	NUCB2, PFKL, SQRDL	–	–	NDUFB4, DLST, ECHS1, IDH	CYB5A, COX1, PTGS1ADH5, ALDH18A1, GLUD1, GLUD2, OGDH, SOD2	–
Cell Adhesion	C1QBP, DDX6, ICAM1, TGFB1I1	STAT1	–	DPP4, ITGA2, IDH1, KTN1	PRKAR1A, PPP3CA, RAC1	PRKCA
Metabolic Processing	APOL2, NUCB2, PFKL	SLC25A1	RANBP2	DLST, ECHS1, IDH1, NDUFB4	ADH5, ALDH18A1, GLUD1, GLUD2, OGDH	CYB5A, COX1, PGP, PTGS1, PRKCA
Post-Translational Modifications	C1QBP, DDX6	–	–	ADAR	SRRT	PRKCA

WJ-MSCs, Wharton’s jelly-derived mesenchymal stem cells; ASFs, Adult skin fibroblasts; NFFs, Neonate foreskin fibroblasts.

**Table 3 ijms-21-06437-t003:** Protein levels predicted by MS analysis, mRNA expression by real time quantitative reverse transcription polymerase chain reaction (qRT-PCR), and protein levels detected by Western blot analysis. − means un-detected; + means detected.

Gene Symbol	Protein Name	Protein Expression Levels Determined by MS	mRNA Levels Determined by qRT-PCR	Protein Levels Determined by Western Blot
WJ-MSCs	NFFs	ASFs	WJ-MSCs	NFFs	ASFs	WJ-MSCs	NFFs	ASFs
*EPHA2*	EPH receptor A2	+	−	−	+++	+	−	+++	+	+
*SLC25A4*	ADP/ATP translocase 1	+	−	−	++	−	−	+++	−	−
*TLN2*	Talin2	+	−	+	−	+	+++	−	−	+++
*LMF2*	Lipase maturation factor 2	−	+	+	−	++	++	+	+	++
*ITGA2*	CD49b/Integrin subunit alpha 2	+	+	−	++	+	−	+++	++	−
*VDAC3*	Voltage-dependent anion channel 3	+	−	−	+	+	+	+++	+	+++
*SOD2*	Superoxide desmutase	+	−	+	+	−	−	+++	−	−
*CDH2*	CD325/N-Cadherin	−	+	−	−	+++	−			
*ITGA5*	CD49e/Integrin subunit alpha 5	++	+	+	++	+	+	++	+	+
*IGF2BP3*	Insulin-like growth factor 2 mRNA binding protein 3	−	−	+	++	+	++			
*PLEC*	Plectin	−	+	+	+	+	+	++	++	++
*CLE7*	RNA transcription, translation and transport factor	+	−	−	+	−	−	+++	+	+
*CDH5*	CD144/VE-cadherin				++	+	−	++	+	+
*NEXN*	Nexilin	+	−	−	++	+	−	+	+	+
*MLCK1*	Myosin light chain kinase	+	−	−

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
