# Peer review of "Comparative Proteomic Analysis Identifies EphA2 as a Specific Cell Surface Marker for Wharton’s Jelly-Derived Mesenchymal Stem Cells"

_ijms, 2020, doi:10.3390/ijms21176437_

Round 1

Reviewer 1 Report

The manuscript by Madhoun and colleagues reports a proteomics study where they identified ephrin type-A receptor 2 (EphA2) as a candidate surface-specific protein for WJ-MSCs. The authors then went on and characterized this and other markers using distinct techniques. In general, the study seems to be well conducted and sound, but previous reports published in the Taiwanese Journal of Obstetrics & Gynecology in 2015 and 2018 (doi 10.1016/j.tjog.2015.10.012; doi 10.1016/j.tjog.2018.08.020) had already identified EphA2 as a marker of MSCs obtained from placenta and umbilical cord, in contrast with fibroblasts (using a proteomics approach validate with other techniques). Hence, the main novelty of the submitted manuscript is seriously compromised.

Specific issues:

  1. In line 53 authors state (referring to MSCs) that ‘As multipotent cells, they can be differentiated in vitro into a wide spectrum of cell types from the three-germ layers [11-14]’. It would be advisable that authors somehow refrain or contextualize better this statement, due to the fact that although MSCs can differentiate into cells that express markers typical of the 3 germ layers, many times such cells fail to be functional. Although MSCs are able to differentiate into osteocytes, adipocytes and chondrocytes, there are limits to their plasticity, hence the term multipotent and not pluripotent, like for example pluripotent stem cells. Some of the earlier studies claiming such high potency for MSCs failed to be reproduced by others.

  1. In line 60, authors state that ‘Unlike MSCs, WJ-MSCs can modulate immunological responses rendering them an immunoprivileged status(…)’. I believe that this statement is inaccurate, since BM-MSCs also display such immune-related capacity. Moreover, the reference cited does not seem to back up the statement. Please clarify.

  1. Data to produce figure 3 seems to be from 3 independent experiments (2 technical duplicates of 3 biological samples). How many ns were considered for the statistical analysis (it is known that the number of ns has an impact on the statistical power)? Only 3 should be considered, not 6. Please clarify.

  1. In figure 4, why are the lanes separated from another? Minimally edited WBs should be shown.

  1. In the Discussion section, authors state (line 240) ‘On the other hand, MSCs were shown to express the fibroblast-specific surface markers, such as collagen, vimentin, fibroblast surface protein, heat shock protein 47, and α-smooth muscle actin [2, 25].’ Certainly not all of the referred proteins are surface markers. Please revise.

  1. Unlike stated by the authors (lines 263-266), references 33 and 34 (the works mentioned in the beginning of this review) identified EphA2 as WJ/UC-MSC marker and not only as a placental MSC marker.

Minor comments

  1. In line 91, perhaps it would be advisable to use the term ‘cell types’ instead of ‘cell lines’.

  1. For being so extensive, Table 1 could go into supplementary data.

  1. In line 366, 'impermeabilized' should be 'permeabilized'?

Author Response

Specific issues:

  1. In line 53 authors state (referring to MSCs) that ‘As multipotent cells, they can be differentiated in vitro into a wide spectrum of cell types from the three-germ layers [11-14]’. It would be advisable that authors somehow refrain or contextualize better this statement, due to the fact that although MSCs can differentiate into cells that express markers typical of the 3 germ layers, many times such cells fail to be functional. Although MSCs are able to differentiate into osteocytes, adipocytes and chondrocytes, there are limits to their plasticity, hence the term multipotent and not pluripotent, like for example pluripotent stem cells. Some of the earlier studies claiming such high potency for MSCs failed to be reproduced by others.

We would like to thank the reviewer. We improved the introduction (lines 48-64). 

  1. In line 60, authors state that ‘Unlike MSCs, WJ-MSCs can modulate immunological responses rendering them an immunoprivileged status(…)’. I believe that this statement is inaccurate, since BM-MSCs also display such immune-related capacity. Moreover, the reference cited does not seem to back up the statement. Please clarify.

We would like to thank the reviewer. We improved the introduction and added a paragraph for the immunomodulatory effect of MSCs from different sources (lines 72-88).

  1. Data to produce figure 3 seems to be from 3 independent experiments (2 technical duplicates of 3 biological samples). How many ns were considered for the statistical analysis (it is known that the number of ns has an impact on the statistical power)? Only 3 should be considered, not 6. Please clarify.

We would like to thank the reviewer. For each cell type, the reverse transcription was done on mRNA extracted from 3 biological samples. The biological samples are from different passages ranging between P2-P5. For each sample, the q-PCR was done in duplicate to reduce pipetting error, average RQ were taken. Then, the mean ± SD for q-PCR from each biological replicate was calculated and shown in Figure 3.    

  1. In figure 4, why are the lanes separated from another? Minimally edited WBs should be shown.

We would like to thank the reviewer. These experiments done at different loading arrangement. For clarification, Example for the original Western blots are shown below. WBs were done several times for each protein. The best image/sample that reflects the calculated average was used in the manuscript. The variations seen in protein expression are most likely related to the passage number and the loading. 

  1. In the Discussion section, authors state (line 240) ‘On the other hand, MSCs were shown to express the fibroblast-specific surface markers, such as collagen, vimentin, fibroblast surface protein, heat shock protein 47, and α-smooth muscle actin [2, 25].’ Certainly not all of the referred proteins are surface markers. Please revise.

We would like to thank the reviewer. We re-wrote the discussion section.

  1. Unlike stated by the authors (lines 263-266), references 33 and 34 (the works mentioned in the beginning of this review) identified EphA2 as WJ/UC-MSC marker and not only as a placental MSC marker.

We would like to thank the reviewer. After Thorough inspection of the study by Shen and his colleagues (reference 33 and 34), it does not define the source of umbilical cord MSCs, although it implies for the perivascular cells or WJ-MSCs:

  1. The abstract does not state the use of umbilical cord WJ-MSCs.
  2. No definition for the umbilical cord cell source (WJ, blood or perivascular tissues).

With the assumption that WJ-MSCs were used in reference 33 and 34, our data confirms these finding and add extra information. In our study, we compared protein profiles in WJ-MSCs, adult fibroblasts and neonatal fibroblasts, the latter cell type is most likely the main contaminant in WJ-MSCs preparations. Furthermore, in our study, we used commercially available cell lines, that are presumably pure preparations, with no prospective contamination with other cell type.

Minor comments

  1. In line 91, perhaps it would be advisable to use the term ‘cell types’ instead of ‘cell lines’.

We would like to thank the reviewer. We re-wrote the paragraph for improvements requested by Reviewer-2. We hope it is accepted to your as well.

  1. For being so extensive, Table 1 could go into supplementary data.

We would like to thank the reviewer. We believe that Table 1 and 2 contain informative data for the readers. If applicable, we suggest showing the tables in a landscape orientation. If not applicable, the editor may like to move both tables to supplementary data as suggested by the reviewer. 

  1. In line 366, 'impermeabilized' should be 'permeabilized'?

We would like to thank the reviewer. The mistake was corrected line 528

Reviewer 2 Report

The authors of the work undertook a very important task, which is determining the universal marker cell WJ-MSC. However, the work requires clarification and a broader description of the methods used in its implementation.

Introduction:

Authors mention only about the Bone Marrow MSC (BN-MSC) as a commonly known MSCs, what about the Adipose Derrived (AD-MSC), which are very easy to obtain and commonly used in many of medicine branches.

In lines 60-61 authors suggest that only WJ-MSC have immunomodulatory potential what is not true. Please change this sentence and compare the immunomodulatory properties of AD-MSC, BM-MSC and WJ-MSC.

I suggest strengthening the message of the importance of undertaking research, which is not emphasized in the introduction.

Methods:

1. The authors do not write about from how many wharton gellys (n number) cells were isolated. In the paragraph "Preparation of protein extract for MS analysis" they mention about three cell lines, is that mean that this is the n of WJ-MSC or other cel lines? If the n=3 in each type of cells per group, in my opinion this is to small for comprehensive statistical analysis.

2. How was the RNA rated? What was the RIN for samples?

3. Western blot and immunofluorescence analysies should be described better, even shortly. qRT-PCR should be also shortly described.

4. In the "Statistical analyses" paragraph, authors describe that all experiments were done in triplicates, but still, if the n was 3, were the normality tests done? I would suggest increase the n numer.

Discussion:

The discussion contains many repetitions of analysis descriptions that should be included in the results. the discussion seems to be artificially divided into small parts, so it's hard to follow. I would ssuggest to enrich it with more literature and strengthen the importance of EphA2 as a surface marker for WJ-MSC.

Figures:

The description of the figures does not indicate the significance for asterixes or other markings.

Author Response

The authors of the work undertook a very important task, which is determining the universal marker cell WJ-MSC. However, the work requires clarification and a broader description of the methods used in its implementation.

We would like to thank the reviewer for his/her impact. We believe that your comments are constructive and has improved our manuscript. Thank you.

Introduction:

Authors mention only about the Bone Marrow MSC (BN-MSC) as a commonly known MSCs, what about the Adipose Derrived (AD-MSC), which are very easy to obtain and commonly used in many of medicine branches.

We would like to thank the reviewer. We re-wrote the introduction with the description of other MSCs sources including BM- and AD-MSCs (lines 46-47, Yellow, bold).

In lines 60-61 authors suggest that only WJ-MSC have immunomodulatory potential what is not true. Please change this sentence and compare the immunomodulatory properties of AD-MSC, BM-MSC and WJ-MSC.

We would like to thank the reviewer. We added a paragraph (lines 72-88, Yellow, bold). Describing the immunomodulatory effect of MSCs from different recourses

I suggest strengthening the message of the importance of undertaking research, which is not emphasized in the introduction. 

We would like to thank the reviewer. We improved the last paragraph emphasizing the message (lines 102-104, Yellow, bold).

Methods:

  1. The authors do not write about from how many wharton gellys (n number) cells were isolated. In the paragraph "Preparation of protein extract for MS analysis" they mention about three cell lines, is that mean that this is the n of WJ-MSC or other cel lines? If the n=3 in each type of cells per group, in my opinion this is to small for comprehensive statistical analysis.

For proteomic analysis, we utilized three replicates from three different passages range between passage 2 to 5) for each cell type i.e. n=9. Corrected in the Materials and Methods section, lines 357-358, Yellow, bold.

  1. How was the RNA rated? What was the RIN for samples?

Total RNA was quantified using NanoDrop 2000c spectrophotometer (Thermo Scientific, USA) and RNA integrity was evaluated using 2% agarose gel electrophoresis (data not shown). Corrected in the Materials and Methods section, lines 423-424, yellow, bold.

  1. Western blot and immunofluorescence analysies should be described better, even shortly. qRT-PCR should be also shortly described.

As requested, we added more details for the Western Blot and qRT-PCR, with a focus on references citations. If we do further short, the methods will lose the significant. Corrected in the Materials and Methods section, lines 404-412, yellow, bold.

  1. In the "Statistical analyses" paragraph, authors describe that all experiments were done in triplicates, but still, if the n was 3, were the normality tests done? I would suggest increase the n numer.

We would like to thank the reviewer. We meant by triplicate: Each experiment was done from three biological samples; however, each was done in a double technical replicates, i.e. n=6. Corrected in the Materials and Methods section, lines 394-395.  

Discussion:

The discussion contains many repetitions of analysis descriptions that should be included in the results. the discussion seems to be artificially divided into small parts, so it's hard to follow. I would ssuggest to enrich it with more literature and strengthen the importance of EphA2 as a surface marker for WJ-MSC.

We would like to thank the reviewer. As suggested, we deleted the sentences related to the results section. In addition, we re-wrote the discussion section for further calcifications and more literature. Focusing on the identified protein and their prospective role.  

Figures:

The description of the figures does not indicate the significance for asterixes or other markings

We would like to thank the reviewer. We added the values to the figure lagends. *P<0.05, **P<0.01 and ***P<0.001 are significant to the higher bar. #P<0.01 is significant to all other bars. Corrected for all figures.